**Data Availability Statement:** All relevant data are within the paper and its Supporting Information files.

**Funding:** The author(s) received no specific funding for this work.

# Self-efficacy and career intention: Uncovering the underlying processes

Md. Sohel Chowdhury[1], Md. Shakhawat Hossain[1], Md. Alamgir Hossain[2], Renhong Wu[3]*

1 Department of Management Studies, University of Barishal, Barishal, Bangladesh, 2 Department of Management, Hajee Mohammad Danesh Science and Technology University, Dinajpur, Bangladesh, 3 School of Management, Kyung Hee University, Seoul, Republic of Korea

* wurenhongbini@163.com

## Abstract

This study aims to suggest and empirically evaluate a conceptual framework that investigates the association of career self-efficacy with career intentions, emphasizing on the specific and sequential mediating mechanisms of two psychological constructs: person-environment (P-E) fit and career attitudes. Relationships among the constructs were hypothesized based on the relevant literature and were examined with AMOS and SPSS Process Macro using survey data collected from a sample of 341 job seekers in Bangladesh. Results showed that P-E fit and career attitudes each partially mediated the relationship between career self-efficacy and career intentions. Noticeably, both the mediators, together in sequence, completely mediated the above relationship. In light of the ensuring chain mediating role of P-E fit and career attitudes, a thorough understanding of the effects of self-efficacy on career intentions not only contributes to the existing literature, but also provides notable implications for career counselors and HR managers in today's competing firms.

## Introduction

Because extant research suggests that failing to attract the right candidates might harm a firm's competitive edge, many scholars have urged HR professionals to boost prospective applicants' psychological attraction to a firm. In this respect, a substantial body of research has applied Ajzen's [1] theory of planned behavior (TPB) to investigate the relationship between job seekers' career self-efficacy and their intentions to engage in a career (hereafter referred to as 'career intentions'). However, despite having inconsistent findings for the above relationship, adequate empirical study on this theorized phenomenon in career literature is still inadequate to date. In addition, while many scholars have demonstrated the influence of fit perceptions on current employees' performance and commitment to their organizations, fit research for potential employees in a prehire recruitment context is very limited [2, 3]. This suggests that the effects of fit perceptions are mainly attributable to a firm's current workers, leaving a significant gap in the field. Another noticeable feature broadly ignored in the previous research is that most studies have concentrated largely on Western settings and empirical research on the

**Competing interests:** The authors have declared that no competing interests exist.

application of a theory-based framework for career choice is scarce in South Asian cultures, and much more so in Bangladesh.

To fill in the gaps in the literature, we suggest a theoretical model to study underlying processes that appear to be salient in the linkage between self-efficacy and career intentions. Drawing on fit perceptions at the individual level [4], we argue that person-environment (P-E) fit, defined as the extent to which job seekers perceive that they are fit for the working environment of a potential recruiting firm [5], and attitudes toward engaging in a career (hereafter denoted as 'career attitudes') can serve as individual and serial mediators for pulling the instrumental side of career self-efficacy so that job seekers' competency beliefs act to align their career intentions. In this study, we used P-E fit perceptions and career attitudes as motivational mediators that can explain why self-efficacy is made manifest in a positive influence on career intentions. Although past research distinctly noted the effects of those psychological variables on applicants' career decisions, none of the research attempted to explain the aforementioned effects in a unified framework, overlooking the reasonable associations among one another. Failing to comprehend the causal mechanisms for the connection between career intentions and self–efficacy may lead to a limited potential candidate pool, and as a result, a firm's edge may be lost to its rivals [6].

Therefore, this study makes some notable contributions to the related area in the following respects. First, this research investigates how job seekers' competency beliefs turn out to be more influential. Examining the underlying procedure of how potential applicants' career self-efficacy beliefs influence their intentions to engage in a career through more proximal effects of psychological variables contributes to the existing literature on career-related research, responding to the findings reported in early studies [e.g., 7, 8]. Second, by probing the sequential mediating mechanism of P-E fit and career attitudes, this research adds value to the theoretical knowledge in this field that has so far not manifestly established clear routes from career self-efficacy to career intentions among millennial job seekers in South Asian culture, much less in the Bangladeshi context. Third, this study combines and builds on two distinct lines of research in career literature: (1) research that tested the impact of self-efficacy on career intentions, but failed to acknowledge the mechanism through which the effect might be exerted [e.g., 9, 10], and (2) research that demonstrated the association between P-E fit perceptions and career intentions, but ignored the causal process through which such a relationship might take place [e.g., 11, 12].

All in all, this study seeks to expand previous research by exploring the sequential multiple mediating effects of P-E fit perceptions and career attitudes in a prehire context to bridge the gap between potential applicants' career self-efficacy beliefs on the one hand and their career intentions on the other. More specifically, this paper suggests and empirically tests a research model for explaining millennial job seekers' occupational intentions to engage in the Bangladesh Civil Service (BCS), one of the largest government employment sources in the country. Despite the fact that the BCS offers a lucrative compensation package for its staff members, job seekers in Bangladesh appear to be more selective to choose a profession as their potential career paths.

## Literature review and hypotheses development

We reviewed the career literature here in order to establish the conceptual framework and hypotheses for this study, which we believe are more useful when exploring the mediating roles of P-E fit perception and career attitudes for understanding the rapport between career self-efficacy and career intentions. Fig 1 represents the research model for this study. The

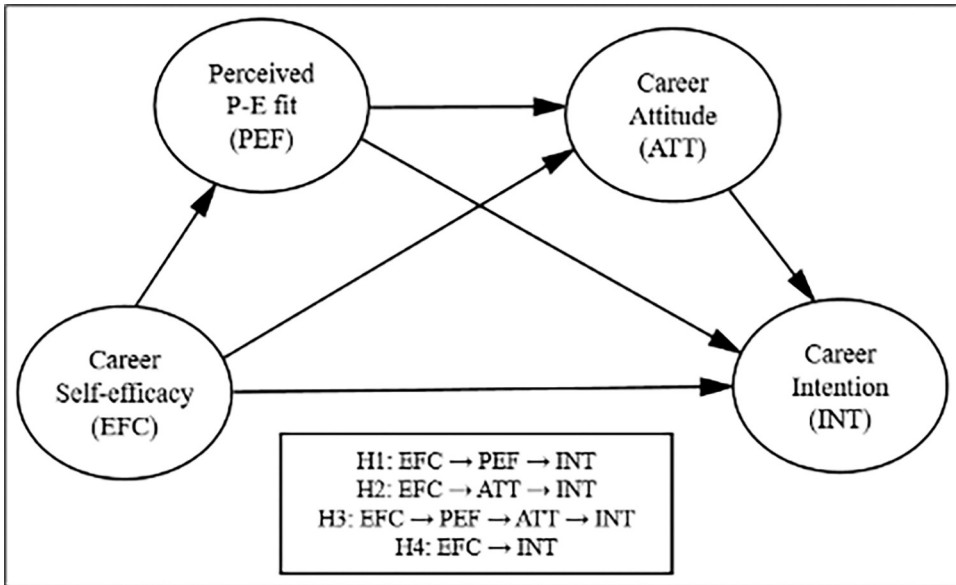

**Fig 1. Hypothesized research model.**

empirical examination of our research model is likely to offer some important implications for career counselors and HR practitioners in today's competing firms.

Intentions to engage in a career result from a process by which individuals make sense of their occupational choices [13, 14]. Finding a preferred career in an organization is important for a person's career development and is particularly critical for unemployed individuals. As noted before, scholars remain interested in career research to predict applicants' intentions to be involved in a career based on various psychological theories. However, one of the most dominant psychological theories in this field is the TPB, which was developed by Ajzen [1]. The TPB posits that a person's behavioral intention is determined by the TPB core constructs: attitudes, subjective norms, and self-efficacy [15, 16]. A person's emotional views over an object or action are referred to as attitudes [17]. Subjective norms are the social pressures that a person perceives to perform a particular action [18]. Self-efficacy denotes individuals' levels of confidence or belief that they possess the required skills and abilities to perform a particular action [19]. Of the variables included in the TPB, self-efficacy was the last variable added to the model by Ajzen [20].

Through the lens of the TPB, extant research on career literature has consistently demonstrated a positive association of attitudes and subjective norms with intentions. Conversely, examinations of the relationship between career self-efficacy and career intentions have yielded inconsistent findings. For instance, while Amani and Mkumbo [21] and Van Hoye et al. [22] found a significant link between self-efficacy and career intentions, Johnston et al. [23] and Yizhong et al. [8] reported that self-efficacy did not significantly predict career intentions. These inconsistencies suggest that there may be some other variables in play between career intentions and self–efficacy. In this paper, we argue that job seekers' P-E fit perceptions and career attitudes are salient psychological factors that may help us to better understand why job seekers exert different career intentions as the outcome of their self-efficacy beliefs. Because of the compelling importance of P-E fit perceptions and career attitudes in this field [24], this study examines their roles in career decision process, specifically concerning the association between job seekers' self-efficacy and intentions related to a career.

P-E fit perceptions depend on individuals' own assessment of how well they conform to the environmental characteristics at work. Along with the TPB core variables, P-E fit perception is also an important antecedent for predicting individuals' intentions and behaviors in various settings, such as career choice [25]. In a meta-analysis, Uggerslev et al. [26] stated that perceived fit is one of the strongest predictors of organizational attractiveness throughout the recruitment process of a firm. In a more recent study, Dutta [12] also noted that P-E fit is more salient in influencing career intentions at the early stage of the recruitment process of an organization. While the majority of the past research regarding P-E fit has focused on existing employees (i.e., post-entry context), relatively few studies to date have considered prospective applicants' pre-entry P-E fit perception and its influence on their career-related decisions. When applied to the present study context, P-E fit refers to prospective employees' judgement about the degree of compatibility between them and the work environment of an organization in which they intend to develop their careers.

## Mediating role of P-E Fit perceptions

Self-efficacy research on career literature shows that prospective employees' career intentions will be optimized when they have the necessary skills and abilities for a career [27]. Past research suggests that self-efficacy not only refers to a person's skills and abilities, but indicates his or her confidence to cope with the environment in question as well [28]. It is unlikely for job seekers to want to have a career in an organization if they do not have the required level of skills and abilities. Furthermore, job seekers who view themselves as competent for a career in an organization are more likely to be intrinsically motivated to aim at the career if they perceive compatibility between them and the work environment. Given that career self-efficacy is an indicator of one's career adaptability [29], perceived P-E fit and its subsequent outcome (i.e., career intentions) may be optimized when job seekers believe that they are able to cope with the work environment of a potential recruiting firm. That is, we argue that job seekers' career self-efficacy is likely to affect their P-E fit perceptions, which in turn consequently lead to their career intentions. Thus, we developed the first research hypothesis as follows:

**H1**. P-E fit perceptions mediate the relationship between career self-efficacy and career intentions.

## Mediating role of career attitudes

A substantial body of TPB research has documented a positive relationship between career attitudes and career intentions [e.g., 8], as well as a positive linkage between job seekers' self-efficacy and their career intentions [e.g., 30]. However, the role of career attitudes in the association between self-efficacy and career intentions to date remains unexplored in this field. To address this phenomenon, we argue that self-efficacy affects career attitudes, which in turn, consequently lead to career intentions. The rationale behind our argument was on the basis of idea that there is a linear association between cognitive constructs and affective or attitudinal constructs [31]. Self-efficacy is related to job seekers' beliefs or convictions which are cognitive in nature, whereas career attitude is an affective construct. Thus, higher self-efficacy beliefs should be associated with higher career attitudes for prospective employees. Additionally, these increased career attitudes, in turn, are likely to influence career intentions as job seekers intend to choose a career if they consider engaging in the career to be a worthwhile activity [22]. On these grounds, we hypothesized that:

**H2**. Career attitudes mediate the relationship between career self-efficacy and career intention.

## Sequential mediating role of P-E Fit and career attitudes

So far, the theoretical underpinnings of this study suggest that prospective job applicants' self-efficacy influences their perceptions of P-E fit, which in turn influences their career attitudes and, eventually, their career intentions. Chowdhury and Kang [32] noted that a multi-stage underlying psychological process is salient for predicting prospective employees' career intentions. Thus, under the research paradigm of this study, it is essential to evaluate the indirect impact of job seekers' self-efficacy on their career intentions via the sequential influence of their P-E fit perceptions and career attitudes. Another reason for investigating the chain mediating effect is that if a conceptual framework depicts an association between A and D as linearly related via B followed by C, then the indirect relationship between A and D via the periodic influence of B and C may be substantial [33]. Hence, it seems plausible to suggest and test the sequential mediation framework so as to precisely realize the impact of linear relations between job seekers' self-efficacy, perceived P-E fit, and career attitudes on their intentions to engage in a career. Thus, we came up with the hypothesis as follows:

**H3**. P-E fit perceptions and career attitudes sequentially mediate the relationship between career self-efficacy and career intentions.

Furthermore, we anticipated that the analysis of hypothesis H3 would be somewhat mediated, given a substantial body of empirical research has established a strong correlation between job seekers' self-efficacy and their career intentions based on the TPB [e.g., 34]. In addition, Di Fabio and Rosen [35] argued that individuals feel interests in a specific activity when they perceive themselves to be efficacious and expect favorable outcomes. Past studies conducted in collective societies such as China, Korea, and Japan have also confirmed the positive effects of career self-efficacy on career intentions [e.g., 27]. Research has demonstrated Bangladesh as a collectivistic country. Therefore, the conceptualization of career self-efficacy and the tendency of job seekers in Bangladesh led us to develop and test the hypothesis H4 to procreate the results of previous studies.

**H4.** Job seekers' career self-efficacy is positively related to their career intentions.

## Research method

### Participants and procedure

A self-reported questionnaire was designed to gather data from job seekers in Bangladesh for empirical assessment of the research hypotheses of this study. As this study did not involve any patient or live vertebrates, an ethics committee was not required to undertake an ethical evaluation. In order to evaluate the usability and the simplicity of comprehension of measuring items in the research context, a pretest with a sample of 20 respondents was performed prior to administering the formal survey. One single respondent raised concern about the mandatory response requirement for all the measurement items on the questionnaire. In fact, one of the benefits of using an online survey approach is that it eliminates the possibility of missing values in the data set. As a result, we were unable to take the argument into account. The measurement items appeared to be operational in the study context. After the pilot study, a link to the online survey form was sent to the respective coordinators of career guiding centers (coaching centers in Bangladesh), requesting them to share the link with enrolled job seekers to complete the form.

Because we employed an online questionnaire survey, according to the survey platform's policy, participation in the survey was voluntary, and respondents' completion of the survey

would signify their consent to participate in the study. The survey questionnaire was completed by 402 respondents in total from October 5, 2023 to November 5, 2023. During the cleansing of data, the standard deviation for the 37 participants' responses to all measurement items was found to be zero, and 24 observations were found to be incomplete. In total, 61 observations were not examined for further analysis in order to prevent possible results of confusion. The remaining sample size (i.e., n = 341) is sufficient for structural equation modeling of the primary four study constructs in social science research [36]. The demographic statistics revealed that most of the respondents were male (66.9 percent), ranged from 21 to 26 years of age (83.6 percent), and were at the graduate level (85 percent). It is worth mentioning that the target sample (i.e., actual job seekers) of this study was a suitable area of investigation as the career intention indicators may possibly differ for actual job seekers than for those who are not yet in the labor market [37].

## Measures

The measurement items for study variables were selected based on the scales that were documented as reliable and valid in prior research. Items were somewhat tailored to fit into the study context. Unless otherwise specified, participants were asked to rate how strongly they agreed or disagreed with the statements on a 5-point Likert-type scale ranging from 1 (strongly disagree) to 5 (strongly agree). The mean value of each construct's items was used to calculate the respective construct. For every case, higher scores reflect higher levels of a construct. The following is a short explanation of the measuring items chosen for each study construct:

**Career self-efficacy.** In this study, we operationalized career self-efficacy as job seekers' beliefs in their abilities to successfully engage in a specific career [38]. For this construct, five items derived from Huang [7] and Arnold et al. [39] were used. One of the items included was "Overall, I am confident that I could easily perform well in the BCS if I wanted to". Cronbach's alpha for this construct was .83.

**Career attitudes.** This construct was measured on a five-point semantic differential scale, where respondents were required to indicate whether they thought working for the BCS was: Bad-Good; Unpleasant-Pleasant; Worthless-Valuable; Harmful-Beneficial; and Unenjoyable-Enjoyable. These bipolar adjectives were demonstrated in many TPB studies [e.g., 7, 39]. The reliability measure for this construct was .88.

**P-E fit perceptions.** Perceived P-E fit was measured using six items adapted from Cable and DeRue [5], and Saks and Ashforth [40]. We considered subjective fit as it has been reported to be superior to any other fit measures [41]. A representative item of perceived P-E fit was "I feel that the BCS will enable me to do the kind of work what I want to do". The reliability estimate of this construct was .91.

**Career intentions.** Job seekers' career intentions were measured using a four-item scale based on Huang [7] and Arnold et al. [39]. We somewhat tailored those items to fit into the study context and the target intention (i.e., to engage in the BCS). A representative item was "As soon as possible, I really intend to be engaged in the BCS". The alpha reliability coefficient of this construct was .93.

**Control variables.** In addition to the psychological constructs discussed above, the respondents' demographic information was taken into consideration. Respondents' age, gender, level of education, and job familiarity (i.e., familiarity with the BCS) were controlled because they were reported to have an influence on career-related decision making in previous research [42].

## Overview of analyses

Consistent with many previous investigations in this field, statistical analyses were carried out in three consecutive phases. First, common method variance (CMV), reliability, validity, and multicollinearity of the study constructs were assessed using confirmatory factor analysis (CFA) with AMOS (version 24). Second, well-established model evaluation metrics were used to determine whether the measurement model and path model accurately characterized the data. Finally, the path model's standardized estimates were used to test the stated hypotheses. Following the procedure of Chowdhury and Kang [32], we employed Hayes's [33] SPSS Process Macro (version 3.4) to evaluate the robustness of the serial mediation effect of P-E fit perceptions and career attitudes.

# Results

## Test of CMV, reliability, validity, and multicollinearity

To confirm whether the measurement instruments were free from the influence of CMV, we incorporated a common latent factor (CLF) for all the survey items as suggested by Podsakoff et al. [43]. The results revealed that none of the factor loadings of the measurement items was affected by the CLF as changes in beta coefficients were not higher than the common threshold value of .20 (shown in Table 1), confirming that CMV was not a major issue in this study. Composite reliability (CR) and Cronbach's alpha ($\alpha$) were taken into account to examine construct reliability. The CR and α value for each construct exceeded the common threshold value of .70 (shown in Table 1), affirming the construct reliability of the study. We examined convergent validity and discriminant validity to confirm construct validity. Table 1 shows that the average value extracted (AVE) and the item loadings ranged from .51 to .79 and .54 to .94 respectively, ensuring the convergent validity. The square root of the AVE of every variable exceeded its correlations with other constructs. In addition, each construct's maximum shared variance (MSV) was lower than its AVE, signifying support for the discriminant validity [44]. Table 1 also shows that the study constructs were free from the multicollinearity problem as all tolerance values exceeded .10, and none of the variance inflation factors (VIF) surpassed the standard threshold value of 10, [36].

## Test of model fit

Before testing the study hypotheses, we examined Hu and Bentler's [45] cutoff criteria for model fit indices such as s Chi-square ($\chi^2$), the ratio of $\chi^2$ to the degree of freedom ($\chi^2/df$), the root mean square error of approximation (RMSEA), the standardized root mean square residual (SRMR), comparative fit index (CFI), and normative fit index (NFI) to ensure whether the measurement model characterized the data well. The fit indices exposed that the hypothesized four-factor model represented the data well ($\chi^2$ = 463.13; $\chi^2/df$ = 2.82; CFI = .94; SRMR = .04; RMSEA = .07; NFI = .92) compared to the one-factor model ($\chi^2$ = 1026; $\chi^2/df$ = 6.00; CFI = .84; SRMR = .08; RMSEA = .12; NFI = .82). Because the measurement model fit the study data well, we included hypothesized links among the latent constructs to test the research model's usefulness. In the structural model, control variables were also included as covariates. A collection of fit indices that were utilized for testing the measurement model was taken into account to determine the structural model fit. The results showed that all the fit indices for the structural model fit the data well ($\chi^2$ = 13.34; $\chi^2/df$ = 1.11; CFI = .99; SRMR = .04; RMSEA = .01; NFI = .98) and the model explained 75 percent of variance in career intentions, indicating explanatory power of the model. As shown in Table 2, significant correlations between the conceptual constructs indicated preliminary support for the research hypotheses.

**Table 1. Reliability, validity, multicollinearity, and CMV statistics.**

| Constructs and Items | CFA Loadings | | CR | AVE | MSV | Cronbach's Alpha (α) | Multicollinearity | |
|---|---|---|---|---|---|---|---|---|
| | d[1] | d[2] | | | | | Tolerance | VIF |
| Career self-efficacy | | | .83 | .51 | .46 | .83 | .52 | 1.89 |
| EFC1 | .75 | .17 | | | | | | |
| EFC2 | .65 | .20 | | | | | | |
| EFC3 | .73 | .17 | | | | | | |
| EFC4 | .54 | .18 | | | | | | |
| EFC5 | .85 | .19 | | | | | | |
| Career attitudes | | | .89 | .63 | .52 | .88 | .28 | 3.52 |
| ATT1 | .84 | .05 | | | | | | |
| ATT2 | .85 | .02 | | | | | | |
| ATT3 | .59 | .04 | | | | | | |
| ATT4 | .85 | .05 | | | | | | |
| ATT5 | .80 | .06 | | | | | | |
| P-E Fit perceptions | | | .91 | .64 | .51 | .91 | .35 | 2.82 |
| PEF1 | .80 | .06 | | | | | | |
| PEF2 | .74 | .08 | | | | | | |
| PEF3 | .74 | .08 | | | | | | |
| PEF4 | .90 | .04 | | | | | | |
| PEF5 | .84 | .05 | | | | | | |
| PEF6 | .77 | .06 | | | | | | |
| Career intentions | | | .93 | .79 | .52 | .93 | | |
| INT1 | .95 | .08 | | | | | | |
| INT2 | .94 | .06 | | | | | | |
| INT3 | .85 | .05 | | | | | | |
| INT4 | .81 | .03 | | | | | | |

Notes. EFC = Career self-efficacy; ATT = Career attitudes; PEF = P-E fit perceptions; INT = Career intentions; d[1] = Standardized CFA loadings; d[2] = CMV effects on individual items; All d[1] CFA loadings are significant at ***$p < 0.001$.

## Hypotheses testing

We explicitly evaluated the relevance of indirect effects of self-efficacy on career intentions through the influence of P-E fit perceptions and career attitudes within our research model, which is consistent with many other studies in this field [e.g., 46]. The analysis was carried out with Model 6 and 5000 bootstrap samples of 95 percent bias-corrected confidence intervals (CI), as suggested by Hayes's [33] SPSS Process Macro. The total and direct effect models showing standardized path estimates are depicted in Fig 2. The specific indirect effects of self-efficacy on career intentions are presented in Table 3. None of the demographic variables was found to have any significant effect on career intentions. As evident from Table 3, the specific indirect effects of self-efficacy on career intentions through P-E fit perceptions and career attitudes were significant, providing support for the hypotheses H1 and H2. Further, the hypothesis H3 was supported as the indirect effect of self-efficacy on career intentions through the sequential influence of P-E fit perceptions and career attitudes was found to be significant (shown in Table 3). As depicted in Fig 2, the total effect model revealed a significant association of self-efficacy with career intentions ($\beta = .60, p < .001$). However, the direct effect model

**Table 2. Means, standard deviations, and correlations among the variables.**

| Variables | M | SD | 1 | 2 | 3 | 4 | 5 | 6 | 7 | 8 |
|---|---|---|---|---|---|---|---|---|---|---|
| 1. Gender[a] | .33 | .47 | | | | | | | | |
| 2. Age[b] | .16 | .37 | .07 | | | | | | | |
| 3. Education[c] | .85 | .35 | -.07 | .09 | | | | | | |
| 4. Job familiarity | .97 | .16 | -.00 | .02 | .13 | | | | | |
| 5. Career self-efficacy | 3.59 | .73 | -.02 | .14** | .03 | .01 | (.71) | | | |
| 6. Career attitudes | 3.50 | .90 | .01 | .11* | .01 | -.03 | .68** | (.79) | | |
| 7. P-E fit perceptions | 3.81 | .92 | .04 | .09 | .01 | .01 | .58** | .70** | (.80) | |
| 8. Career intentions | 3.68 | 1.15 | -.00 | .03 | .04 | .03 | .59** | .72** | .70** | (.89) |

Notes.

[a]Coded 0 = Male, 1 = Female

[b]Coded 0 = 21–26, 1 = 27–32 (in years)

[c]Coded 0 = Undergraduate, 1 = Graduate

**p < 0.01

* p < 0.05;; The square root of the corresponding construct's AVE is shown in the brackets.

shows that self-efficacy could not predict career intentions ($\beta$ = .04, $p$ > .05) after the inclusion of P-E fit perceptions and career attitudes in the model, indicating full mediation. Therefore, no support was found for the hypothesis H4.

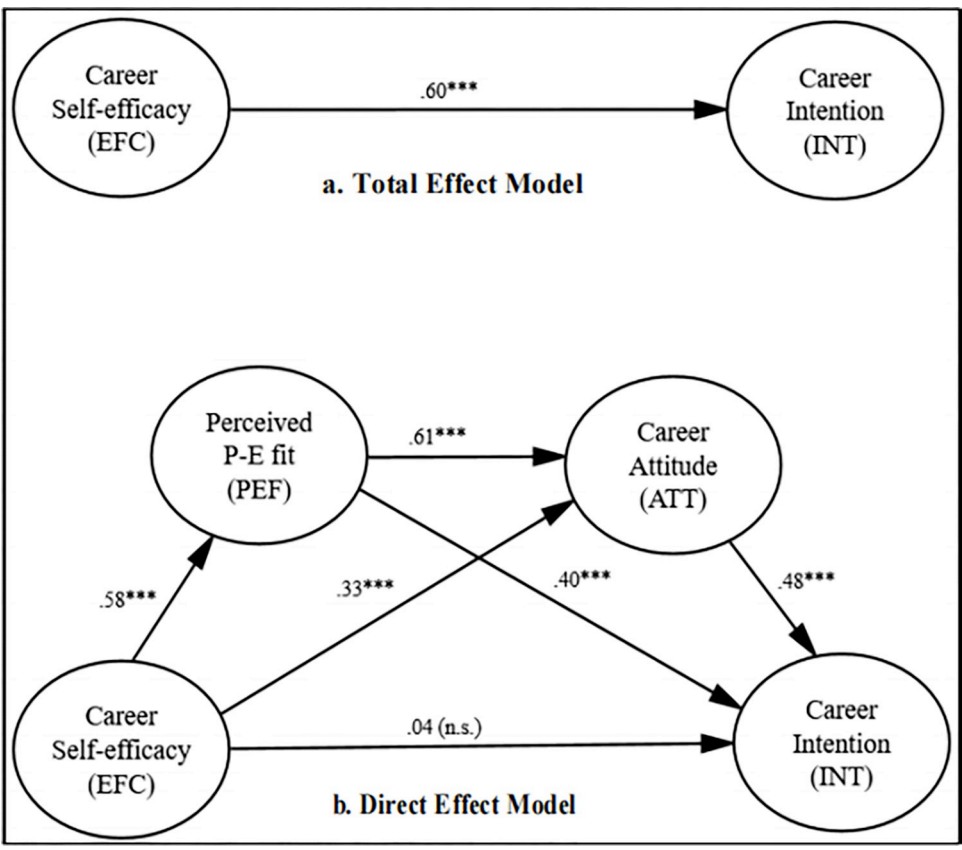

**Fig 2. Standardized coefficients of sequential mediation model.**

**Table 3. Direct, indirect, and total effects of career self-efficacy on career intentions.**

| Effects | Path Estimates (β) | SE | BC 95% CI | | Hypothesis (Supported/ Rejected) |
|---|---|---|---|---|---|
| | | | Lower | Upper | |
| Indirect (H1): EFC→PEF→INT | .58 × .40 = .23 | .04 | .16 | .32 | Supported |
| Indirect (H2): EFC→ATT→INT | .33 × .48 = .16 | .03 | .10 | .22 | Supported |
| Indirect (H3): EFC→PEF→ATT→ INT | .58 ×. 61 × .48 = .17 | .02 | .12 | .22 | Supported |
| Total indirect effects of EFC on INT | .56 | .04 | .47 | .65 | |
| Direct effect (H4): EFC→INT | .04 | .06 | -.06 | .17 | Rejected |
| Total effects of EFC on INT | .60 | .07 | .81 | 1.08 | |

Notes. EFC = Career self-efficacy; ATT = Career attitudes; PEF = P-E fit perceptions; INT = Career intentions; SE = Standard error; BC = Bias corrected;

CI = Confidence intervals.

## Discussion

The main purpose of the study was to examine the psychological mechanisms underlying the relationship between job seekers' career competency beliefs and intentions to engage in a career. We uncovered evidence for a theoretical model suggesting that P-E fit perceptions and career attitudes individually and sequentially mediate the linkage of job seekers' self-efficacy with their career intentions. The study findings are congruent with past research which convincingly noted strong association of P-E fit perceptions with career attitudes [e.g., 47], career intentions [e.g., 11], and of career attitudes with career intentions [e.g., 9]. However, our study also contributes to the existing literature by demonstrating chain mediation effects of P-E fit perceptions and career attitudes, where self-efficacy first improves P-E fit perceptions, which in turn enhance career attitudes that consequently lead to career intentions. It is worth mentioning that, apart from the sequential mediation path, two simple mediation paths with either P-E fit perceptions or career attitudes as a single mediator for self-efficacy–career intentions link were also found to be significant. These findings further corroborate the salient aspects of both of these self-related constructs for career intentions and highlight the dynamic ways in which self-efficacy acts on individuals' aspirations for paid employment.

Contrary to the hypothesis, but consistent with many prior studies [8,e.g., 23], the direct effect of self-efficacy on career intentions was found to be insignificant in our research model. It appears that the mediating effects of P-E fit perceptions and career attitudes are robust for the interpretation of career-related decisions, such as finding careers in an organization. Another possible explanation for why self-efficacy could not significantly predict career intentions might be due to the measurement scale, which could have led to lower variation in results. It should be emphasized that when the attitudinal construct was removed from our study model, the results revealed partial mediation as the direct effect of self-efficacy on career intentions was found to be significant ($\beta = .19$, $p < .001$), suggesting that both the mediators together caused the insignificant direct relationship between job seekers' self-efficacy beliefs and career intentions in our proposed model. This disparity in findings highlights the need for examination of P-E fit and career attitudes as the sole mediator for the above relationship. Thus, along with career attitudes, the inclusion of perceived P-E fit in the model offers additional insights into the multifaceted nature of relationships among the variables under consideration in this study.

### Theoretical contributions

This study contributes to the literature in the following ways. First, our study provides an understanding of the salient underlying process through which job seekers' self-efficacy shapes

their intentions to engage in a career. Despite the fact that previous research documented a significant influence of fit perceptions and career attitudes on career choice separately, none of those researches, in relation to self-efficacy, considered them in a unified framework. This study has shown how job seekers' self-efficacy becomes contributory to predicting their intentions to engage in a career through more proximal outcomes of P-E fit perceptions and attitudes toward the career. Therefore, this study could work as an impetus for further research on such underlying mechanisms, via which prospective employees' self-efficacy affects their career intentions.

Second, a key finding of this study was that the direct impact of perceived P-E fit on career intentions was found to be influential within our research model, refuting the notion that any construct outside the TPB model can only affect the intention construct indirectly through the influence of any of the TPB core constructs, such as self-efficacy and career attitudes in the study [48]. Moreover, P-E fit perceptions individually played a mediating role in the linkage of self-efficacy with career intentions. Noticeably, while career attitudes influenced career intentions, self-efficacy, as a TPB core construct, could not play a direct role in predicting career intentions in our research model. This divergence in results sheds light on the complexities of multiple mediating mechanisms in the association between self-efficacy and career intentions.

Third, by incorporating P-E fit literature into the TPB context, this study provides insight into how psychological constructs can work together to elucidate the associations between job seekers' self-efficacy and career intentions. By demonstrating the chain mediating effects of P-E fit perceptions and career attitudes, this research contributes to the theoretical advancement of career literature that, to date, has not convincingly provided clear paths from career self-efficacy to career intentions. Furthermore, although previous research has mostly focused on the Western context, less is known about job seekers' career-related decision-making process in terms of their efficacy beliefs in developing nations, particularly Bangladesh. As a result, this research is critical in determining the generalizability of the research findings.

## Managerial implications

This research also provides some useful implications for career counselors and HR practitioners in today's competing firms. First, it would not be wise to consider job seekers' competency beliefs as the only direct antecedent of their intentions to engage in a career. According to the study findings, potential employees' P-E fit perceptions determined by their self-efficacy beliefs tended to affect their career attitudes, which might be successfully undertaken as an important determinant for influencing their intentions to engage in a career. In addition, our work has confirmed that job seekers' attitudes toward a career are directly influenced by their self-efficacy beliefs. Therefore, career education programs focusing on the nature and competency requirements of a career should be provided to counselors, who in turn should share this knowledge with job seekers to help improve their self-efficacy beliefs and P-E fit perceptions, as well as to prepare them with more career-related information.

Second, the study findings revealed that job seekers' P-E fit perceptions affect their career intentions, supporting the notion that individuals form their perceptions based on the social information they receive and process [49]. For instance, a job seeker who obtains information from others working in a company that encourages its employees to generate new ideas might provide a sense that the company' environment prefers creativity at work. As job seekers lack adequate access to company information [50], they tend to get information from social sources to assess the compatibility between their competency beliefs and working environment of a potentially recruiting firm, which may work as a key checkpoint for their career-related decisions. Therefore, it is recommended that HR practitioners may consider their company

websites, social media platforms, or other recruitment channels to communicate its working environment with potential human resources.

Third, millennial job seekers are not merely attracted to a career by the compensation and benefits that it offers to them [32]. Rather, given that individuals' behavioral intentions are a function of their perceptions about the reality [51], HR managers should be concerned with job seekers' P-E fit perceptions in addition to compensation and benefits. Our study provides HR managers with insights into how job seekers' career attitudes and intentions are determined by their P-E fit perceptions. It is expected that a firm develops and executes plans and policies to make the working environment convenient for its employees. However, during the applicant attraction phase of its recruitment campaign, it might not be less important to share information about its working atmosphere with potential employees to influence their intentions to pursue a career in the firm, as suggested by the study findings.

### Limitations and future research directions

There were certain limitations to this study that can be used to guide future research. First, considering the research objectives, this study did not focus on the other TPB core construct (i.e., subjective norms) and the actual career choice behaviors of prospective employees. Given that a person's intention to perform certain actions may not always result in actual performance [52], the inclusion of a behavioral construct as the ultimate dependent variable in our research model could produce different outcomes. Therefore, it is recommended that future studies examine the full TPB model by incorporating subjective norms and actual career choice behaviors to provide a better understanding of the intention–behavior linkage.

Second, consistent with many past studies in this field, we considered global scales to measure job seekers' career self-efficacy. However, Creed et al. [53] suggested that self-efficacy could be classified further as a measure of task self-efficacy and controllability. In a similar vein, Jung et al. [27] demonstrated generalized self-efficacy and job search self-efficacy as different constructs in their study. Therefore, to elucidate the possible role of self-efficacy in predicting career intentions, future follow-up studies may investigate the direct and indirect effects that different types of self-efficacy measures might have on job seekers' occupational intentions.

Last but not least, as this study was based on a cross-sectional design, reverse causality could play a role in predicting outcomes [54]. Consistent with previous research, we controlled the respondents' age, gender, education, and job familiarity. However, a career could be pursued for other reasons also. Furthermore, since our sample was made up entirely of Bangladeshi job seekers, readers should exercise caution when extrapolating the results. As a consequence, we suggest that future studies use a longitudinal design to target job seekers from various cultural settings in order to generate more solid findings that can be generalized.

### Supporting information

**S1 File. Original dataset.** This file contains the original dataset used in the analysis. (XLSX)

**S2 File. Inclusivity-in-global-research-questionnaire.** (DOCX)

### Author Contributions

**Conceptualization:** Md. Sohel Chowdhury, Renhong Wu.

**Formal analysis:** Md. Sohel Chowdhury.

**Investigation:** Md. Shakhawat Hossain.

**Methodology:** Md. Shakhawat Hossain.

**Software:** Md. Shakhawat Hossain.

**Validation:** Renhong Wu.

**Visualization:** Md. Shakhawat Hossain, Md. Alamgir Hossain, Renhong Wu.

**Writing – original draft:** Md. Sohel Chowdhury, Md. Shakhawat Hossain.

**Writing – review & editing:** Md. Alamgir Hossain, Renhong Wu.

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
