## [Decision Letter · Decision Letter 0]

27 Sep 2024

PONE-D-24-25408Self-efficacy and Career Intention: Uncovering the Underlying ProcessesPLOS ONE

Dear Dr. Wu,

Thank you for submitting your manuscript to PLOS ONE. After careful consideration, we feel that it has merit but does not fully meet PLOS ONE’s publication criteria as it currently stands. Therefore, we invite you to submit a revised version of the manuscript that addresses the points raised during the review process.

We look forward to receiving your revised manuscript.

Kind regards,

Bo Pu, Ph.D.

Academic Editor

PLOS ONE

4. In the online submission form you indicate that your data is not available for proprietary reasons and have provided a contact point for accessing this data. Please note that your current contact point is a co-author on this manuscript. According to our Data Policy, the contact point must not be an author on the manuscript and must be an institutional contact, ideally not an individual. Please revise your data statement to a non-author institutional point of contact, such as a data access or ethics committee, and send this to us via return email. Please also include contact information for the third party organization, and please include the full citation of where the data can be found.

5. Please ensure that you include a title page within your main document. You should list all authors and all affiliations as per our author instructions and clearly indicate the corresponding author.

Additional Editor Comments:

this manuscript should be major revised.

Reviewers' comments:

Reviewer's Responses to Questions

**Comments to the Author**

1. Is the manuscript technically sound, and do the data support the conclusions?

Reviewer #1: No

Reviewer #2: Yes

2. Has the statistical analysis been performed appropriately and rigorously? 

Reviewer #1: Yes

Reviewer #2: Yes

3. Have the authors made all data underlying the findings in their manuscript fully available?

Reviewer #1: Yes

Reviewer #2: Yes

4. Is the manuscript presented in an intelligible fashion and written in standard English?

Reviewer #1: No

Reviewer #2: No

5. Review Comments to the Author

Reviewer #1: Can you provide more detailed definitions of the key constructs (career self-efficacy, P-E fit, career attitudes, and career intentions) to ensure clarity for readers who may not be familiar with these concepts?

How was the sample of 341 job seekers in Bangladesh selected, and does this sample adequately represent the broader population? Could you discuss any potential sampling biases and how they might impact the generalizability of the findings?

Could you elaborate on the specific procedures used in AMOS and SPSS Process Macro? For instance, how were the mediation effects tested, and were any assumptions made during the analysis that could influence the results?

How were the measurement tools (e.g., scales for career self-efficacy, P-E fit, and career attitudes) validated in this study? Please attched instrument with details source.

The study suggests that P-E fit and career attitudes sequentially mediate the relationship between career self-efficacy and career intentions. Could you provide a more in-depth discussion on why this sequence was hypothesized and how it aligns with existing literature?

The paper mentions notable implications for career counselors and HR managers. Could you expand on these practical implications, possibly providing examples of how the findings can be applied in real-world settings?

While the study discusses the implications of the findings, could you also elaborate on the limitations of your study and suggest areas for future research to build upon your findings?

Given that the study was conducted in Bangladesh, how might cultural factors influence the relationships among the constructs? Would you consider discussing how these findings could be generalized to other cultural contexts or whether cultural differences might alter the results?

In revision version please provide dataset for check statistical measures.

Reviewer #2: Introduction

Provide a source for the first sentence. Again, the researchers failed to buttress most claims with literature.

While the study draws on the Theory of Planned Behavior (TPB), there is insufficient explanation of how TPB specifically relates to P-E fit and career attitudes in this context. This could leave the reader questioning the theoretical integration.

There is a lack of consistent definitions for some key terms. For instance, "career intentions" and "career attitudes" are mentioned multiple times, but their distinct meanings and interrelationships aren't always clear.

Although the passage mentions the scarcity of research in South Asian contexts, particularly Bangladesh, it does not sufficiently discuss why this cultural setting is significant for understanding the relationship between self-efficacy and career intentions.

The passage does not clearly justify why a theoretical model with sequential mediation is the most appropriate approach for investigating the relationships in question.

The introduction lacks coordination and its difficult to understand what the authors want to achieve. Kindly reorganize your thoughts and improve on it.

Literature Review and Hypotheses Development

There is a misfit here, in the introduction, the researchers provided that TPB has been overly used and so they proposed a theory, however, this theory has not been addressed. Kindly start the argument of the hypothesis development with the theory. Again, is the P-E fit a theory or what? Kindly be specific.

Methods

The researcher did well by stating the time period for data collection, however, don’t you think that the issue the researchers are attending to has been addressed or changed looking at the time duration. What was the target population and what was the actual sample size before distributing the instrument.

Why did the researchers mention control variables in this section but did not discuss their role and implication in the introduction?

Discussion of Results

Kindly discuss the results in lune with the theory.

6. PLOS authors have the option to publish the peer review history of their article (what does this mean?). If published, this will include your full peer review and any attached files.

Reviewer #1: No

Reviewer #2: No

---

## [Author Response · Author response to Decision Letter 0]

26 Oct 2024

We have submitted it in the form of a document.

Thank you very much.

---

## [Decision Letter · Decision Letter 1]

18 Nov 2024

Self-efficacy and Career Intention: Uncovering the Underlying Processes

PONE-D-24-25408R1

Dear Dr. Wu,

We’re pleased to inform you that your manuscript has been judged scientifically suitable for publication and will be formally accepted for publication once it meets all outstanding technical requirements.

Kind regards,

Bo Pu, Ph.D.

Academic Editor

PLOS ONE

Additional Editor Comments (optional):

This manuscript should be published in PLOS ONE.

Reviewers' comments:

Reviewer's Responses to Questions

**Comments to the Author**

1. If the authors have adequately addressed your comments raised in a previous round of review and you feel that this manuscript is now acceptable for publication, you may indicate that here to bypass the “Comments to the Author” section, enter your conflict of interest statement in the “Confidential to Editor” section, and submit your "Accept" recommendation.

Reviewer #2: All comments have been addressed

2. Is the manuscript technically sound, and do the data support the conclusions?

Reviewer #2: Yes

3. Has the statistical analysis been performed appropriately and rigorously? 

Reviewer #2: Yes

4. Have the authors made all data underlying the findings in their manuscript fully available?

Reviewer #2: Yes

5. Is the manuscript presented in an intelligible fashion and written in standard English?

Reviewer #2: Yes

6. Review Comments to the Author

Reviewer #2: the authors have been able to address all issues raised by reviewers and i solely believe that this papers is worth acceptance.

7. PLOS authors have the option to publish the peer review history of their article (what does this mean?). If published, this will include your full peer review and any attached files.

Reviewer #2: No

---

## [Editor Report · Acceptance letter]

19 Dec 2024

PONE-D-24-25408R1 

PLOS ONE

Dear Dr. Wu, 

I'm pleased to inform you that your manuscript has been deemed suitable for publication in PLOS ONE. Congratulations! Your manuscript is now being handed over to our production team.

Kind regards, 

on behalf of

Dr. Bo Pu 

Academic Editor

PLOS ONE